Genome-wide identification and characterization of ABA receptor pyrabactin resistance 1-like protein (PYL) family in oat

Mi Wenbo
Liu Kaiqiang
Liang Guoling
Jia Zhifeng
Ma Xiang
Ju Zeliang
http://orcid.org/0000-0002-9234-2918 Liu Wenhui qhliuwenhui@163.com
Key Laboratory of Superior Forage Germplasm in the Qinghai-Tibetan Plateau, Qinghai Academy of Animal Husbandry and Veterinary Sciences, Qinghai University , Xining , China
Johnson Charles
Electronic publication date: 2023 Oct 2
Publication date: 2023
Volume: 11
Electronic Location ID: e16181
Received 2023 Apr 14; Accepted 2023 Sep 5
Copyright: © 2023 Mi et al.
Copyright year: 2023
Copyright holder: Mi et al.
License: This is an open access article distributed under the terms of the Creative Commons Attribution License, which permits unrestricted use, distribution, reproduction and adaptation in any medium and for any purpose provided that it is properly attributed. For attribution, the original author(s), title, publication source (PeerJ) and either DOI or URL of the article must be cited.
License URL: https://creativecommons.org/licenses/by/4.0/

Keywords: Avena sativa L., PYL family, Genome-wide, Expression patterns

Funding: Key Laboratory of Superior Forage Germplasm in the Qinghai–Tibetan Plateau 2020–ZJ–Y03 China Agriculture Research System CARS-34 This work was supported by the Key Laboratory of Superior Forage Germplasm in the Qinghai–Tibetan Plateau (2020–ZJ–Y03) and the China Agriculture Research System (CARS-34). The funders had no role in study design, data collection and analysis, decision to publish, or preparation of the manuscript.

==============================
Abscisic acid (ABA) is a phytohormone that plays an important role in plant growth and development. Meanwhile, ABA also plays a key role in the plant response to abiotic stressors such as drought and high salinity. The pyrabactin resistance 1-like (PYR/PYL) protein family of ABA receptors is involved in the initial step of ABA signal transduction. However, no systematic studies of the PYL family in “Avena sativa, a genus Avena in the grass family Poaceae,” have been conducted to date. Thus, in this study, we performed a genome-wide screening to identify PYL genes in oat and characterized their responses to drought stress. A total of 12 AsPYL genes distributed on nine chromosomes were identified. The phylogenetic analysis divided these AsPYLs into three subfamilies, based on structural and functional similarities. Gene and motif structure analysis of AsPYLs revealed that members of each subfamily share similar gene and motif structure. Segmental duplication appears to be the driving force for the expansion of PYLs, Furthermore, stress-responsive AsPYLs were detected through RNA-seq analysis. The qRT-PCR analysis of 10 AsPYL genes under drought, salt, and ABA stress revealed that AsPYL genes play an important role in stress response. These data provide a reference for further studies on the oat PYL gene family and its function.

Introduction

Oat (Avena sativa L.), one of the most important cereal crops grown worldwide, is prized as one of the richest sources of vitamin B1, plant-based protein, and healthy fat (Gutierrez-Gonzalez, Tu & Garvin, 2013; Mahmoud et al., 2022). In addition, oats play a prominent role in livestock production due to their variety, high yield, and excellent nutritional quality (He & Bjornstad, 2012). Oats are primarily produced in the temperate regions of the northern hemisphere, especially in Canada, Russia, the United States, Germany, Australia, Finland, and China. However, environmental stressors such as drought can severely constrain oat yields (Tajti, Pál & Janda, 2021). Thus, breeding oat varieties with increased drought resistance is of considerable importance and schemes to improve the yield and quality of oat will require the efficient screening of beneficial genes. Water availability is a limiting factor for many crop plants (Seleiman et al., 2021). Drought stress negatively impacts both plant development and yield by dysregulating important physiological and biochemical processes (Fahad et al., 2017). Under drought stress, signal molecules detect and transmit stress signals to activate the expression and regulation of a variety of stress-related genes. This signaling cascade results in a series of adaptive physiological and biochemical changes in order to maintain basal metabolism and dynamic balance with the external environment (Neill et al., 2008). Abscisic acid (ABA), a phytohormone and key signaling molecule, extensively regulates plant growth, development, and stress response, and the ABA content increases rapidly upon exposure to drought, saline-alkali, and other stress conditions (Zhu, 2016). Under abiotic stress, ABA signaling is mediated by the PYL-PP2C-SnRK2 core signaling pathway. The PYL protein family interacts with other proteins, such as phosphatases in the APP2C branch. Several members of the PYL family have been characterized in plants (Dittrich et al., 2019; Park et al., 2009). Structural analyses have defined the ternary crystal features of PYLs, the PP2C proteins, and their substrate ABA (Meskiene, 2004; Santiago et al., 2012).

As a key component of ABA signaling, the specific functions of PYL proteins have been studied in a variety of plants. In Arabidopsis, AtPYL13 and AtPYL6 have been found to restrain seed germination (Fuchs et al., 2014), and AtPYR1 and AtPYL8 have been found to facilitate ABA-mediated root growth, seed germination, and stomatal closure (Gonzalez-Guzman et al., 2012; Zhao et al., 2014). Overexpression of AtPYL4 increased the drought tolerance of Arabidopsis (Pizzio et al., 2013). Moreover, AtPYL6 was found to interact with MYC2, a core jasmonic acid (JA)-responsive transcription factor, effectively linking the ABA and JA signaling pathways to regulate a series of downstream developmental and stress response processes (Aleman et al., 2016). In rice, OsPYL5, OsPYL9, and OsPYL10 receptors have been characterized through overexpression, and PYL-knockout mutants exhibit enhanced abiotic stress tolerance and yield (Chen et al., 2017; Kim et al., 2014; Verma et al., 2019). In maize, Overexpression of ZmPYL8, ZmPYL9, and ZmPYL12 resulted in increased cold tolerance (He et al., 2018). Taken together, these studies indicate that the PYL gene family plays an important role in both plant development and abiotic stress response. Identifying the PYL genes in oat will be essential to not only better understand ABA signal transduction in this important crop, but also to breed oat varieties with enhanced stress resistance.

Genetically, oat is an allohexaploid species (2n = 6x = 42), with three (A, C, and D) homoeologous subgenomes. These characteristics make oat an ideal model for studies of polyploidization, subfunctionalization, and homoeologous chromosome interactions (Peng et al., 2022). Members of the PYL gene family have been successfully identified in many plants, including 14 PYLs in Arabidopsis (Dittrich et al., 2019), 13 PYLs in rice (Yadav et al., 2020), 38 PYLs in wheat (Lei et al., 2021), 14 PYLs in tomato (Sun et al., 2011), and 29 PYLs in tobacco (Bai et al., 2019). However, the PYL gene family has not been fully characterized in oat, hampering its application in oat stress improvement. In this article, we identified members of the PYL gene family in oat and analyzed the role of these AsPYLs in drought resistance. The results of this study will provide a theoretical basis for the development of oat varieties resistant to drought.

Materials and Methods

Identification of the PYL genes in Avena sativa

The oat HMM file and proteins sequence of the PYL domain (PF10604) were obtained from the grain genes database (https://wheat.pw.usda.gov/) (Wang, Wang & Paterson, 2012) and Pfam database (https://ebi.ac.uk/Tools/pfa/pfamscan/) (Finn et al., 2000), respectively. According to the HMMER file to search, the PYL genes in the oat proteins. All candidate genes were further confirmed by the Pfam, CDD (https://www.ncbi.nlm.nih.gov/cdd/) (Lu et al., 2019), and SMART (http://smart.embl-heidelberg.de/) (Jrg et al., 2000). Then, we also studied the Mw and pI of all found genes by Expasy (https://web.expasy.org/compute_pi/) (Panu et al., 2012). Furthermore, the subcellular localization of these genes was also identified by Cell-PLoc (http://www.csbio.sjtu.edu.cn/bioinf/Cell-PLoc-2/) (Horton et al., 2007).

Phylogenic and conservative motif analysis of PYL family members

Multiple sequence alignments of the AsPYL protein domain were performed using MEGA 7. Then, phylogenetic reconstruction based on the PYL domain of Arabidopsis, maize, wheat, rice and oat were obtained using the neighbor-joining method, and the program parameters were as follows: p-distance, 80% cutoff of partial deletion, and 1,000 bootstrap repeats. The conserved motifs of all AsPYL proteins were studied using MEME (https://meme-suite.org/meme/) (Bailey et al., 2009), and the default parameters were as follows: the number of motifs was 10, with the optimum motif width ranging from 6 to 50. The exon-intron structure of the AsPYL genes were acquired from the genome annotation file using TBtools software (Chen et al., 2020). The upstream 2,000 bp sequences of all AsPYL genes were screened using PlantCare (https://bioinformatics.psb.ugent.be/webtools/plantcare/html/) to identify the cis-elements of the promoter (Lescot, 2002).

Chromosomal location and synteny analysis of PYL genes

According to the oat genome annotation file, TBtools shows chromosome distribution and mapping of all AsPYL genes (Chen et al., 2020). The multicollinearity Scan toolkit (MCScanX) with default parameters was used to analyze the gene repetition events (Wang et al., 2012). To show the synteny of orthologous PYL genes acquired from oat with maize, Arabidopsis, rice, and wheat, a synteny analysis plot was constructed using TBtools (Dual Synteny Plotter).

Expression analysis of AsPYL under various conditions

Expression data of AsPYL under stress and drought conditions was calculated based on previous study (Liu et al., 2022; Wu et al., 2017). Moreover, this studied analyzed transcriptome data under various stress conditions. Gene expression and heat maps were performed using TBtools software (Chen et al., 2020).

Protein interaction network of AsPYLs

The protein interaction network of AsPYLs were predicted by GeneMAINA (http://genemania.org/) and String (https://www.string-db.org/; medium confidence 0.400), based on maize orthologous proteins. The protein–protein interaction (PPI) network and node network diagrams were constructed using the Cytoscape software 3.2 (Institute for Systems Biology, Seattle, WA, USA) (Shannon, 2003).

Plant material, growing conditions, drought and salt treatments

Oat seeds were sown in the soil in plastic tanks and grown in greenhouses. After 2 weeks, the seedlings were acclimated to growth chamber conditions for 48 h and further treated in 200 mM NaCl solutions, 10% PEG6000, and 100 μM ABA for 0, 2, 4, 8, 12, and 24 h, respectively. Then, the treated seedlings were then harvested for RNA extraction and stored at −80 °C.

qRT-PCR analysis

The total RNA was extracted using the plant RNA Kit (Takara, Shiga, Japan), the cDNA was synthesized using a PrimeScript™ RT reagent Kit (Takara, Shiga, Japan). The 10 AsPYL genes were selected to validate the expression level under salt and drought stress and primers, as listed in Supplemental File 1. qRT-PCR was carried out on Light Cycle96 with TB Green® Premix Ex Taq™ II (Takara, Shiga, Japan), the PCR conditions was conducted as follows: 30 s at 95 °C, 40 cycles of 95 °C for 5 s and 60 °C for 40 s. The relative expression levels of AsPYL genes were analyzed by the 2−ΔΔCT method (Livak & Schmittgen, 2013).

Results

Identification of PYL gene family members in Avena sativa

A total of 12 PYL genes were identified in oat, which were distributed in nine out of 21 oat chromosomes (Fig. 1). Chromosomes 1C, 1D, and 6D had the highest abundance of PYL genes (two each), followed by 1A, 4C, 6A, 6C, 7A, and 7D (one each). No PYL genes were identified on chromosomes 2, 3, or 5. In total, 3, 4, and 5 AsPYL genes were identified in the A, C, and D subgenomes, respectively. These results suggest no significant differences in PYL gene abundance at the subgenome scale.

Figure 1 Chromosomal location of AsPYL genes.

The chemical and physical properties of the PYL genes and PYL proteins, including isoelectric point (pI), gene length, open reading frame (ORF) length, exon number, amino acid (aa) length, and molecular weight (MW), were estimated using the ExPASY website. The AsPYLs ranged in length from 201 to 254 aa, with an average of 222 aa. The MW of the AsPYLs ranged from 21.65 to 26.92 kDa, with an average of 24.05 kDa. The predicted pIs of the AsPYLs ranged from 5.07 to 9.90, with an average of 6.29. Eleven of the AsPYL proteins were predicted to be localized to the cytoplasm and chloroplasts, with only one AsPYL extracellularly localized (Table 1).

Table 1 Basic information of AsPYL family genes and their proteins in Avena sativa.

Gene	Gene ID	Gene length
(bp)	ORF length
(bp)	No. of
exon	Predicted protein	Subcellular
localization	
Size (aa)	MW (kDa)	pI	
AsPYL1	AVESA.00001b.r3.1Ag0000971.2	1,109	738	1	245	25.89	5.59	Chloroplast	
AsPYL2	AVESA.00001b.r3.1Cg0000613.2	2,045	618	3	205	21.97	5.60	cytosol	
AsPYL3	AVESA.00001b.r3.1Cg0001670.2	914	723	1	240	25.39	5.31	Chloroplast	
AsPYL4	AVESA.00001b.r3.1Dg0000157.1	2,030	618	3	205	21.97	5.61	cytosol	
AsPYL5	AVESA.00001b.r3.1Dg0000962.2	1,386	738	1	245	25.78	5.47	Chloroplast	
AsPYL6	AVESA.00001b.r3.4Cg0003243.2	1,274	765	1	254	26.41	8.2	Chloroplast	
AsPYL7	AVESA.00001b.r3.6Ag0000500.3	3,461	726	2	241	26.92	9.9	cytosol	
AsPYL8	AVESA.00001b.r3.6Cg0002306.1	3,545	642	3	213	23.71	6.26	cytosol	
AsPYL9	AVESA.00001b.r3.6Dg0000148.1	961	609	1	202	21.65	5.07	cytosol	
AsPYL10	AVESA.00001b.r3.6Dg0000219.1	3,665	642	3	213	23.79	5.93	cytosol	
AsPYL11	AVESA.00001b.r3.7Ag0002908.1	4,683	606	3	201	22.54	6.45	cytosol	
AsPYL12	AVESA.00001b.r3.7Dg0001928.1	4,756	606	3	201	22.55	6.06	Extracellular	

A phylogenetic tree was constructed based on the 12 AsPYL proteins using MEGA 7 software. According to sequence similarity, the AsPYL proteins can be divided into three subfamilies: I, II, and III (Figs. 2 and 3A). Among the 12 AsPYL proteins, four (AsPYL1, AsPYL3, AsPYL5, and AsPYL9) belong to subfamily I, one (AsPYL6) belongs to subfamily II, and seven (AsPYL2, AsPYL4, AsPYL7, AsPYL8, AsPYL10, AsPYL11, and AsPYL12) belong to subfamily III.

Figure 2 Phylogenetic analysis of PYL gene family in Avena sativa and its progenitors.

Tree was constructed by MEGA7.0 using neighbor-joining method with 1,000 bootstraps. Purple, blue, green, red, and yellow circles represent PYL protein sequences from wheat (Ta), maize (Zm), Arabidopsis (At), rice (Os) and oat (As), respectively. The branch length represents the magnitude of genetic change. Blue, gray and green areas represent subfamilies I, II and III, respectively.

Figure 3 Phylogenetic relationship, conserved motifs and gene structure of AsPYL proteins in Avena sativa.

(A) The phylogenetic tree was constructed based on 12 AsPYL full-length proteins. (B) The conserved motifs, numbers 1–10, are shown in different colored boxes. (C) The exon-intron structure of AsPYL genes. Yellow boxes indicate exons, green boxes indicate untranslated region (UTR) and black lines represent introns.

Comparison to PYL gene family members in other species

To clarify the phylogenetic relationships between oats and other species, a neighbor-joining tree was constructed using the full-length protein sequences of 38 TaPYLs, 13 ZmPYLs, 14 AtPYLs, 13 OsPYLs, and our 12 identified AsPYLs (Fig. 2). Consistent with previous studies in Arabidopsis and wheat, all studied PYLs were found to be classified into three subfamilies. In total, subfamily I contained 22 genes (eight TaPYLs, three ZmPYLs, four AtPYLs, three OsPYLs, and four AsPYLs), subfamily II contained 24 genes (12 TaPYLs, five ZmPYLs, three AtPYLs, three OsPYLs, and one AsPYL), and subfamily III contained 44 genes (18 TaPYLs, five ZmPYLs, seven AtPYLs, seven OsPYLs, and seven AsPYLs). In addition, the AsPYLs were found to be most closely related to the TaPYLs, suggesting a close genetic relationship between oat and wheat.

Gene structure and conserved motifs of AsPYL genes

The conserved motifs of PYL proteins were studied using the MEME program, and 10 motifs were identified across all 12 AsPYL proteins (Fig. 3B). Motifs 1, 2, and 3 were conserved across all PYLs, while motif 4 was conserved among all PYLs except AsPYL6 and AsPYL9. Overall, two AsPYLs (AsPYL 6 and 9) were found to contain three motifs, five AsPYLs (AsPYL2, 4, 7, 11, and 12) were found to contain five motifs, and five AsPYLs (AsPYL1, 3, 5, 8, and 10) were found to contain seven motifs. Although the AsPYLs contained many conserved motifs, each subfamily member was found to also contain unique motifs. These results suggest that while functional similarities exist among members, the presence of unique motifs is indicative of specialized biological functions.

The exon-intron structures of AsPYL genes were also analyzed (Fig. 3C), and all 12 AsPYLs contained either one (AsPYL1, 3, 5, 6, and 9), two (AsPYL7), or three (AsPYL2, 4, 8, 10, 11, and 12) exons. Interestingly, no introns were identified in the subfamily I or II AsPYL genes. All members of subfamily III, except AsPYL7 (one intron), contained two introns. In general, AsPYL genes belonging to the same subfamily tended to share similar exon-intron structures, affirming their close evolutionary relationships.

Synteny analysis of AsPYL genes

The increase in evolutionary raw materials resulting from gene replication can facilitate evolution (Cannon et al., 2004; Kong et al., 2007). Here, we analyzed fragment duplications and tandem duplication events in oats. As shown in Fig. 4, we identified six pairs of segmental duplications: AsPYL4 and AsPYL2, AsPYL5 and AsPYL3, AsPYL5 and AsPYL1, AsPYL1 and AsPYL3, AsPYL7 and AsPYL10, and AsPYL11 and AsPYL12. To further study the evolutionary mechanism among PYL gene family members, we established a collinearity relationship with the four model plants Arabidopsis, rice, maize, and wheat (Fig. 5). The number of syntenic genes between oat and other species was as follows: 25 orthologous pairs between eight AsPYLs and 16 TaPYLs, 13 paired collinearity relationships between 12 AsPYLs and 7 OsPYLs, 11 pairs between 10 AsPYLs and seven ZmPYLs, and two pairs between Arabidopsis and oat. In total, we found 11 common AsPYL genes among these collinearity relationships. In addition, some AsPYLs were found to be associated with the presence of at least two syngenetic pairs (particularly between oat and wheat). For example, two or more homologous pairs contained seven genes (87.50%) in wheat, one (8.33%) in rice, and one (10%) in maize. These genes might play a key evolutionary role.

Figure 4 Synteny analysis of AsPYL genes family in Avena sativa.

The gray line represents the syntenic blocks in the oat genome, and the red lines represent segmental duplication gene pairs. Chromosomes 1–7 are shown in different colors.

Figure 5 Syntenic analysis of PYL gene family among genomes.

Gray lines in the background represent collinear blocks within the genome of Avena sativa and other plants, while red lines highlight represents the collinearity of PYL gene pairs. The species were A. thaliana, Z. mays, O. sativa and T. aestivum respectively.

Structural analysis of AsPYL gene promoters

In order to better understand the transcriptional regulation and function of the AsPYL genes, we predicted the cis-regulatory elements in their promoters. A total of 41 functional cis-elements were identified. The CGTCA-motif, TGACG-motif, G-box, and other key promoter elements were found in all 12 AsPYLs. In addition, a large number of phytohormone-responsive promoter elements were identified among the AsPYLs, as well as those involved in growth and development, signal transduction, and abiotic stress (Fig. 6). Among them, ABA-responsive element (ABRE) and anaerobic-responsive element (ARE) were identified in all 12 AsPYLs. In addition, methyl-jasmonate (MeJA)-responsive motifs (TGACG and CGTCA) were identified in 11 AsPYLs, gibberellin (GA)-responsive motifs (TATC-box and P-box) were identified in eight AsPYLs, light-responsive element (LRE) G-box was identified in 11 AsPYLs, auxin (Aux)-responsive element (TGA) was identified in seven AsPYLs, salicylic acid (SA)-responsive element (TCA) was identified in 5 AsPYLs, and ethylene (ET)-responsive element (ERE) was identified in 8 TaPYLs. Several stress-related elements, including low temperature-responsive (LTR) MYB binding sites and defense- and stress-responsive TC-rich repeats, were also found in AsPYLs. These cis-acting elements assist or act on PYL genes and form PYL-involved plant regulatory networks, and may play key roles in the oat response to abiotic and biotic stressors.

Figure 6 Putative cis-elements existed in the 2 kb upstream region of Avena sativa PYL genes.

(A) The elements which respond are displayed in differently coloured boxes. (B) The number of cis-elements present on the Oat PYL gene.

Gene Ontology and Kyoto Encyclopedia of Genes and Genomes analyses of AsPYL proteins

The functions of AsPYL proteins were predicted by Gene Ontology (GO) analysis. In the molecular functions category, the proteins were found to be associated with protein phosphatase inhibitor activity (GO0004864), molecular function inhibitor activity (GO0140678), protein dimerization activity (GO0046983), and monocarboxylic acid binding (GO0033293). In the cellular components category, the proteins were found to be associated with the nucleus (GO0005634), cytosol (GO0005829), cytoplasm (GO0005737), and intracellular membrane-bounded organelle (GO0043231). In the biological process category, the proteins were found to be associated with negative regulation of protein serine/threonine phosphatase activity (GO1905183), cellular response to alcohol (GO0097306), regulation of phosphatase activity (GO0010921), and negative regulation of phosphatase activity (GO0010923). Kyoto Encyclopedia of Genes and Genomes (KEGG) analysis indicated that the AsPYL proteins were primarily enriched in plant hormone signal transduction, signal transduction and environmental information processing, and MAPK signaling pathway-plant (Supplemental Files 2 and 3).

Protein interaction network of AsPYLs

To elucidate the AsPYL regulatory network, a protein-protein interaction (PPI) network was produced. Overall, 4 AsPYLs had orthologous relationships with ZmPYLs and 10 corresponding interacting functional genes were identified. As expected, most of the proteins interacting with AsPYLs were important and functionally-validated components of the ABA signaling complex, such as PP2C (Fig. 7). Based on functional annotation, the proteins interacting with AsPYLs were divided into four categories, including five phosphatase 2C family proteins (PP2C4, PP2C6, PP2C7, PP2C10, and PP2C11), two protein-serine/threonine phosphatase proteins (Orphan56 and Orphan328), two probable protein phosphatase 2Cs (GRMZM2G045452 and GRMZM2G059453), and one RNA helicase (prh14).

Figure 7 Predicted protein-protein interaction networks of AsPYL proteins with other proteins using STRING tool.

The green circles represent oat PYL proteins, and the circles on the inside represent proteins that interact with AsPYLs. Different colors including dark green, red, yellow, and blue phosphatase 2C family proteins, protein-serine/threonine phosphatase protein, probable protein phosphatase 2C, and RNA helicase, respectively. The two circles connected by the gray line represent the interaction between the proteins.

Expression profiling of AsPYL genes by RNA-seq

Given the broad regulatory role of the PYL family in plants, we studied the expression patterns of all 12 AsPYL genes using publicly-available transcriptome data derived from salt- and drought-stressed oats (Supplemental Files 4 and 5). Notably, not all AsPYL genes responded positively to adversity. For example, AsPYL11 and AsPYL12 were unexpressed under salt and drought stress treatment. Other AsPYL genes were either upregulated or downregulated depending on the treatment, time, and variety. At the same time, the expression levels of different AsPYL genes exhibited differences under the same treatment. For example, AsPYL2, AsPYL4, AsPYL7, AsPYL8, and AsPYL10 exhibited higher expression under salt treatment (Fig. 8A), while AsPYL4, AsPYL7, and AsPYL8 exhibited higher expression under drought treatment (Fig. 8B). These results indicate that AsPYL2, AsPYL4, AsPYL7, AsPYL8, and AsPYL10 may play a significant role in oat drought resistance.

Figure 8 Expression pattern of 12 AsPYL genes in different adversity based on RNA-seq data.

(A) Gene expression analysis of oat under salt stress at 0, 2, 4, 8, 12, 24 h. (B) The comparison of drought stress between DY2 and MW varieties at 0, 6, and 24 h.

Expression patterns of AsPYL genes under abiotic stress

To examine the expression levels of AsPYL genes under drought, salt, and ABA stress, qRT-PCR analysis was performed on 10 AsPYL genes (Fig. 9, Supplemental File 6). We found that the majority of the tested AsPYLs were induced by drought, salt, and/or ABA treatment. For example, the expression levels of AsPYL1/2/4/6/7/8/9/10 were significantly up-regulated under salt and drought stress, and the expression levels of AsPYL3/4/5/7/8/10 were significantly up-regulated under ABA stress. Interestingly, the majority of the up-regulated AsPYL genes (AsPYL2, 4, 7, 8 and 10) belong to subfamily III, that members of the same subfamily tend to share similar expression patterns. Conversely, the expression levels of AsPYL3 and AsPYL5 were inhibited by drought and salt treatment and the expression levels of AsPYL2, AsPYL6, and AsPYL9 were inhibited by ABA treatment.

Figure 9 Expression pattern of 10 selected AsPYL genes in response to salt, drought, and ABA stress.

* indicates significant differences at the 0.05 level, ** indicates extremely significant differences at the 0.01 level.

Discussion

ABA is an important phytohormone that regulates the plant stress response. Simultaneously, ABA plays an important role in mediating important agronomic traits such as seed germination and maturation and physiological responses to abiotic stress (Finkelstein, Gampala & Rock, 2002; Sun et al., 2011; Xiong, Schumaker & Zhu, 2002; Zhu, 2016). Recently, the discovery of the RCAR/PYL family of ABA receptor proteins served as a breakthrough in our understanding of ABA signaling (Park et al., 2009; Ma et al., 2009). Specifically, PYL receptors play an indispensable role in the initiation of ABA signaling pathways. The function of the PYL gene has been characterized in many model species, including Arabidopsis (Dittrich et al., 2019), rice (Yadav et al., 2020), and tobacco (Bai et al., 2019). However, there are few studies on the PYL gene family in oat.

In this study, we identified 12 AsPYL genes and analyzed their basic structure. The genes were named AsPYL1 to AsPYL12 based on their chromosomal locations. Based on phylogenetic analysis, the AsPYL family can be classified into three subfamilies: I, II and III (Figs. 2A and 3). Subfamily I and II AsPYLs are intronless, although all identified subfamily III AsPYLs have introns (Fig. 2C). The intron/exon structure and exon number of AsPYL genes are similar to those in maize (He et al., 2018), wheat (Lei et al., 2021), rice (Yadav et al., 2020), and cotton (Zhang et al., 2017), indicating that the quantity and structure of exons and introns are reflective of phylogenetic relationships (Fig. 3). Introns play an important role in the post-transcriptional regulation of gene expression through both splicing-independent and splicing-dependent intron-mediated mRNA accumulation (Gallegos & Rose, 2019). All of the subfamily III AsPYL members have evolved introns to improve their regulatory activity. In order to study the diversity of the AsPYL proteins, their conserved motifs were identified using the MEME program (Fig. 2B). All 12 AsPYL proteins contained motifs 1, 2, and 3, indicative of the greatly conserved function of this protein family. Phylogenic analysis of AsPYL genes revealed that genes within each subfamily tended to contain the same motifs. For example, motifs 7, 8, and 9 were specific to subfamily I members, with the exception of AsPYL9. Motifs 5, 6, and 10 were endemic to most subfamily III members. Although AsPYL genes of the same subfamily likely have similar biological functions, their specific functions remain to be elucidated. Gene duplication events can be generally divided into five types, including tandem duplication (TD), whole-genome duplication (WGD), proximal duplication (PD), transposed duplication (TRD), and dispersed duplication (DSD), all of which promote gene family expansion in eukaryotes (Doerks et al., 2002; Moore & Purugganan, 2003). WGD events can produce a large number of duplicate genes in a short time (Wang, Wang & Paterson, 2012). WGDs are considered a major feature of eukaryotic genomes, and play an important role in genomic and genetic evolution (Moore & Purugganan, 2003). Several gene families, such as the F-box and SWEET families and heat-shock transcription factors, expanded primarily through WGD and DSD (Li et al., 2016; Qiao et al., 2015; Wang, 2018). However, the expansions of the AP2/ERF and WRKY gene families mainly resulted from TD events (Guo et al., 2014). In this study, we analyzed gene duplication events in oats, and found that six pairs of gene experienced WGD during genetic evolution. Meanwhile, we evaluated a collinearity relationship between Arabidopsis, maize, rice, wheat, and oat (Fig. 5). Collinear gene pairs between oat and maize, rice, and wheat PYL genes were most common on oat chromosomes 1, 6, and 7; maize chromosomes 1 and 5; rice chromosomes 2 and 10; and wheat chromosomes 1, 2, 4, and 7. Some of the PYL genes exhibited multiple collinearity. Further studies on collinearity of oat and wheat showed that most collinear genes belong to subfamily (Additional File 7). These results further demonstrate that the AsPYL genes in subfamilies I and II exhibit higher conservation compared to those in subfamily III.

Cis-acting elements, regulatory motifs which act as transcription factor DNA binding sites, play a core role in gene transcriptional regulation (Kao, Cocciolone & Mccarty, 1996; Pl et al., 2020). When plants experience biotic or abiotic stress, transcription factors are activated and bind to cis-acting elements to promote the expression of related stress-responsive genes (Xing et al., 2018). Furthermore, cis-acting elements are significant regulators of phytohormone responses related to both growth and defense (Zhang et al., 2022). In this study, the AsPYL genes were found to be enriched in light-responsive and phytohormone-responsive (ABA, IAA, GA and SA) modules such as ABRE, ARE, TGACG-motif, CGTCA-motif, and G-box. In particular, ABRE and ARE elements are present in all AsPYL genes. ABRE elements exist in the promoter sequences of many ABA-inducing genes, and play a vital role in salt tolerance and drought resistance (Pla et al., 1993). We infer that the AsPYL genes may take part in the regulation of oat development and defence response. Studies have shown that overexpression of PYL genes improves stomatal regulation, seed germination, seedling growth, and drought resistance in Arabidopsis (Saavedra et al., 2010; Santiago et al., 2009). In rice, the ABA receptor OsPYL has been found to regulate ABA signaling, including improving the sensitivity of seeds to ABA and increasing the drought and salt-alkali stress resistance of rice (Kim et al., 2012; Kim et al., 2014). These results are in agreement with our study, suggesting that AsPYL genes take part in the stress defense response through phytohormone signaling pathways.

To evaluate the responses of AsPYL genes to various stresses, we obtained publicly available transcriptomic qRT-PCR data on drought- and salt-stressed oats. The majority of AsPYL genes were responsive to drought and salt stress, but the expression levels of these genes were not absolutely consistent with the transcriptome data. Therefore, we selected genes involved in drought and salt stress identified in the transcriptome data set for qRT-PCR verification. The results indicate that these genes exhibit high expression levels under drought and salt stress.

Conclusion

In conclusion, 12 AsPYL genes were identified and classified into three subgroups. Their gene structure, phylogenetic relationships, chromosomal locations, conserved motifs, promoter regions, and transcriptomic expression were systematically analyzed. Analysis of AsPYL expression patterns and functions indicated that the majority of AsPYL genes actively participated in growth, development, and stress response. These data provide a reference for further studies on the oat PYL gene family and its function.

Availability of data

The genome sequence of oat is available in https://wheat.pw.usda.gov/GG3/graingenes-downloads/pepsico-oat-ot3098-v2-files-2021. The 12 AsPYL protein sequences and ids are listed in Supplemental Files 8 and 9 and the sequences of PYLs from Arabidopsis (https://www.arabidopsis.org), maize, rice and wheat genomes (https://plants.ensembl.org/index.html) are available in the respective database. The protein sequences of these species are given in Additional File 6.

Supplemental Information

Supplemental Information 1 Supplementary material.

Click here for additional data file.

Additional Information and Declarations

Competing Interests

Author Contributions

Data Availability

The authors declare that they have no competing interests.

Wenbo Mi conceived and designed the experiments, prepared figures and/or tables, and approved the final draft.

Kaiqiang Liu performed the experiments, authored or reviewed drafts of the article, and approved the final draft.

Guoling Liang analyzed the data, prepared figures and/or tables, and approved the final draft.

Zhifeng Jia analyzed the data, prepared figures and/or tables, and approved the final draft.

Xiang Ma analyzed the data, prepared figures and/or tables, and approved the final draft.

Zeliang Ju analyzed the data, prepared figures and/or tables, and approved the final draft.

Wenhui Liu analyzed the data, authored or reviewed drafts of the article, and approved the final draft.

The following information was supplied regarding data availability:

The raw data is available in the Supplemental Files.

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
