# Peer review of "Genome-wide identification and characterization of ABA receptor pyrabactin resistance 1-like protein (PYL) family in oat"

_PeerJ, doi:10.7717/peerj.16181_

## Round 0.1 · original submission · Major Revisions

Please address the issues mentioned by our reviewers, especially those dealing with language, the quality of the phylogenetic tree, the protein expression network, and miRNA target analysis.

·

Basic reporting

1.The methods of Phylogenic and conservative motif analysis should include parameter.

2.The language of all MS should be polished. Such as line 103 “Based on 2.6 x 10-9 substitution/synonymous site in oat (Peng et al. 2022)”, this sentence is problematic.

3.This sentence in line 105 “AsPYL gene expression profiles were calculated under salt stress and drought conditions (Liu et. 106 al. 2022; Wu et al. 2017)” is problematic. It should be “ Expression data of AsPYL under stress and drought conditions was calculated based on previous study (Liu et. 106 al. 2022; Wu et al. 2017) ”.

4.The phylogenetic tree in Figure 7 should contain the bootstrap value. The evolutionary relationships of these genes should refer to the bootstrap values.

Experimental design

no comment

Validity of the findings

no comment

Additional comments

no comment

Reviewer 2 ·

Basic reporting

The text it is not well written. There are a lot of typo mistakes and several sentences in poor english (i.g. line 14, 16, 18, 32, 45, 46 and so on)

The background provided is ok and article structure is correct.

Experimental design

Oat is a nutritional crop with high economic value. On the other hand, abscisic acid (ABA) is a key plant hormone involved in abiotic stress tolerance. In the present manuscript authors have performed a characterization of the ABA receptor family in A. sativa.

Results are relevant to the field. Although AsPYL naming is misleading respect to the
extensive information generated in arabidopsis and other species. Gene naming based on their chromosome localizations have less sense that naming according the phylogenetic relationship and biochemical properties.

The subfamily I in the phylogenetic tree (fig 2) is mixing distant branches, this must be revised.

review citations throughout the text (i.g. line 47, 267, etc)

Explain the meaning of: Gene replication events (TD, WGD, PD, TRD and DSD)... line 289

Validity of the findings

no comment

Reviewer 3 ·

Basic reporting

The authors have done genome wide identification of PYL gene family in oats and analysed its chromosomal location, phylogeny, synteny etc. They have also studied the expression of PYL genes in drought and salt stressed plants in RNA seq data and also by qRTPCR. The authors have further identified miRNA regulation of these PYL genes.

English language needs significant improvement throughout the manuscript.

Experimental design

The work is within the scope of the journal.
1. The miRNA target analysis is not clearly mentioned in the material and methods section. Also this analysis may not be accurate as the authors have used miRNA from many diverse plant species. The threshold and the filtering options have also not been mentioned in the manuscript.
2. Since the PYL genes act like ABA receptors, the authors should have performed experiment to show their induction by ABA.
3. The protein-protein interaction network also needs more explaining.
4. Figure 10 needs improvement. The result part of RNA seq, qRTPCR, PI and miRNA targets need to be written in greater detail.

Validity of the findings

The PYL gene 4, 7, 8, 10 show high expression in drought and salt stresses. The protein interaction network should have been constructed using these genes.
The miRNA study should either be dropped or only Avena sativa miRNA should be used for significant analysis.

---

## Round 0.2 · Minor Revisions

Dear Authors

The previous Academic Editor is no longer available and so I have taken over handling your submission.

According to reviewer 1's comments, the manuscript needs a minor revision to be reconsidered for publication. The authors are invited to revise the paper considering all the suggestions made by the reviewer. Please note that requested changes are required for publication.

Best Regards

·

Basic reporting

1.“Based on 2.6 x 10-9 substitution/synonymous site in oat (Peng et al. 2022)”, this sentence is problematic.This error has not been corrected yet.

2.Comment 4. The phylogenetic tree in Figure 7 should contain the bootstrap value. The evolutionary relationships of these genes should refer to the bootstrap values.
Response 4. Thank you for your suggestion. The phylogenetic tree contain 1000 bootstrap repeats. We have provided explanations in the materials and methods.

1000 bootstrap repeats is not the bootstrap value.I hope the author can thoroughly study the principles of evolutionary trees and make modifications.

Experimental design

no comment

Validity of the findings

no comment

Additional comments

no comment

Reviewer 2 ·

Basic reporting

no comments

Experimental design

no comments

Validity of the findings

no comments

Additional comments

Appropriate changes were made.

---

## Round 0.3 · accepted · Accept

Thank you for revising your publication, congratulations.

In particular, the Section Editor noted:

> I see that in the abstract - authors refer to "oat" as if oat = Avena sativa, but don't mention anything about the genus/species in the abstract. As far as I know, this genus/species is referred to as the common oat, but probably should refer more to the scientific name throughout the manuscript.

·

Basic reporting

This article can be accepted.

Experimental design

no comment

Validity of the findings

no comment

Additional comments

no comment